# Design of a Slotted Waveguide Antenna Based on TE20 Mode in Ku-Band Suitable for Direct Metal Laser Sintering

Youssef Chairi [1,2,3], Sarra Abedrrabba [1], Rozenn Allanic [1], Anne-Charlotte Amiaud [2], Ahmed El Oualkadi [4], Cédric Quendo [1], Thomas Merlet [2], Kamal Reklaoui [3] and Thierry Le Gouguec [1,*]

1  UMR CNRS 6285 Lab-STICC, Université de Brest, 6 Avenue Victor Le Gorgeu CS93837, 29283 Brest, France; youssef.chairi@univ-brest.fr (Y.C.); abderrabbasarra@gmail.com (S.A.); rozenn.allanic@univ-brest.fr (R.A.); cedric.quendo@univ-brest.fr (C.Q.)
2  Thalès LAS/OME, 2, Avenue Gay Lussac, 78990 Elancourt, France; anne-charlotte.amiaud@thalesgroup.com (A.-C.A.); thomas.merlet@thalesgroup.com (T.M.)
3  Faculté des Sciences Tanger, Université Abdelmalek Essâadi Tanger, Tanger BP 416, Morocco; k.reklaoui@uae.ma
4  Laboratoire des Technologies de L'Information et de la Communication (Labtic) Ecole Nationale des Sciences Appliquées, Université Abdelmalek Essâadi Tanger, Tanger BP 1818, Morocco; aeloualkadi@uae.ac.ma
*  Correspondence: thierry.legouguec@univ-brest.fr

**Abstract:** This paper describes the design of a novel Ku-band slotted waveguide antenna (SWA), taking into consideration the advantages and drawbacks of using a 3D direct metal laser sintering (DMLS) process. Indeed, the DMLS process makes it possible to produce a SWA with 64 radiating slots and its feeding network in a single and monolithic process. However, considering the lack of accuracy of the process, the whole design must be completely thought out and planned to avoid sensitive dimensions. Coupling irises inside such structures are elements of major sensitivity in this regard, so the radiating waveguides in the present work were designed to be TE20 ones, avoiding this kind of iris. Thus, a TE10 to TE20 converter was designed to be implemented in the overall power supply structure of an antenna array made up of four linear SWAs with 16 slots each. Both the elementary 16-slot SWA and the complete SWA with the feeding network were manufactured using DMLS. At a resonant frequency of 15 GHz, the measured realized gain is 22.26 dB with sidelobe levels below 10.1 dB. The measured reflection coefficient is lower than −12.6 dB at the center frequency. These measured performances confirm the proof of concept.

**Keywords:** 3D printing; antenna arrays; direct metal laser sintering; feeding network; mode converter; slotted waveguide antenna



## 1. Introduction

Slotted waveguide antenna (SWA) arrays are popular antennas in navigation, radar, and 5G communication networks [1,2]. Compared with planar antenna arrays, SWA arrays have the advantages of high-power handling, low losses, and a higher gain, and thus provide better quality communications. In general, such antennas consist of several parts produced separately by expensive machining techniques, which are then assembled by dip brazing, taking particular care to avoid losses due to misalignment. The development of additive layer manufacturing (ALM) methods, such as stereolithography (SLA) or direct metal laser sintering (DMLS), now offers a broad range of possibilities to realize shapes that could not be made by conventional processes, and thus permits the design of innovative [3,4] and low-cost [5–7] 3D microwave components. Regarding 3D metal printing, the capabilities offered by the DMLS process can be deployed to produce monolithic components in one step with better temperature resistance than SLA associated with a metal plating process [8]. Manufacturing using a single and monolithic approach avoids all faults arising from assembly that are inherent to conventional manufacturing.

However, the DMLS process for microwave components is still under development and roughness and dimension precision inherent to this process have an impact on RF performances [8]. For the case of the SWA array, the RF performances are sensitive to dimensional variations from manufacturing, particularly the size of resonant coupling irises placed between the radiating waveguides and the feeding network [9]. These iris dimensions are difficult to control when using the DMLS manufacturing process [10,11], and no post-process or adjustment is possible inside the structure. The use of the TE20 mode to feed substrate integrated waveguide (SIW) travelling wave antennas or SIW slot arrays has already been described [12–14], but SIW SWA arrays suffer from dielectric losses and greater dispersion than air-filled devices [15].

This paper proposes to take advantage of the monolithic DMLS process, managing the limitations caused by dimension sensitivity through design. Thus, the SWA will be excited using the upper waveguide mode TE20 rather than the classical TE10 and a coupling iris. Adopting this method avoids the use of internal coupling slots [16] by replacing them with a T junction and offers the advantage of allowing compact size and simplified geometry compared with a SWA with TE10 mode propagation [13]. The DMLS process makes this manufacturing approach easy and greatly limits the impact of sensitivity. To demonstrate the concept, this paper proposes to use the DMLS process to manufacture a monolithic SWA array in Ku-band (15 GHz) and its TE20 powering network based, which avoids dielectric losses, presents more thermal stability, and is manufactured in one step.

This paper is organized as follows. In the next Section 2, the design and performances of a TE10 to TE20 mode converter are presented. In Section 3, the use of the TE20 propagation mode to excite a 16 slot SWA array is first presented and then the design of an array of 64 slots with an RF power network distribution working in TE20 mode is presented. The Section 4 of this paper then presents our conclusions and perspectives for future work.

## 2. Rectangular Waveguide TE10 to TE20 Mode Converter

To be able to work with TE20 mode propagation and to make measurements, a TE10 to TE20 mode converter needed to be developed. Waveguide mode converters have already been developed for many microwave applications [17,18]. The TE10 to TE20 mode converter in the present study was developed and designed based on the work reported in [19], considering the central frequency to be 15 GHz. The structure is presented in Figure 1.

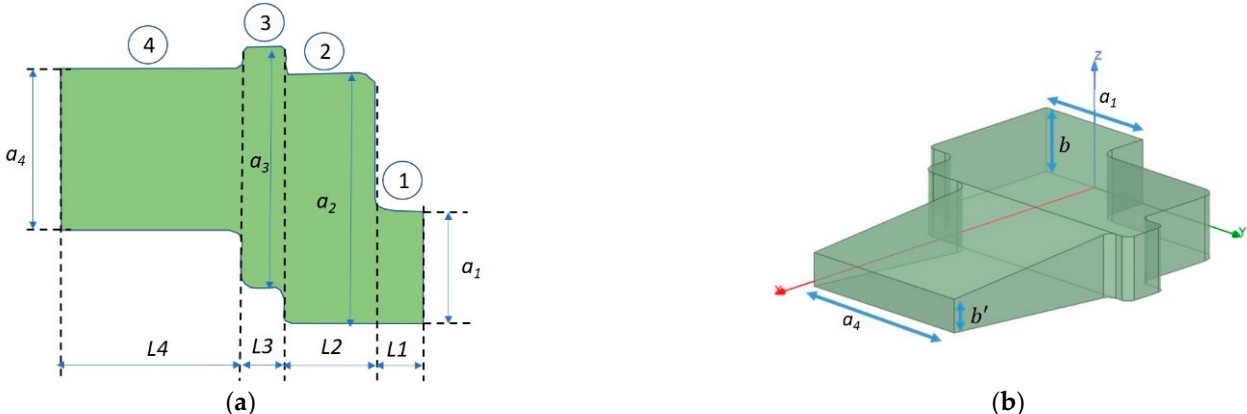

**Figure 1.** Structure of the TE10 to TE20 mode converter: (**a**) upper view with the 4 sections (1–4) of the converter and (**b**) 3D view.

Figure 1a shows an upper view of the converter, highlighting the four longitudinal parts, numbered from 1 to 4. The first part corresponds to a standard WR62 waveguide (a1 = 15.8 mm and b = 7.9 mm) that supports the TE10 mode in the Ku-band and will serve for conventional excitation and measurements. The use of an eccentric connection between this WR62 guide and an oversized waveguide allows us to excite the TE30 mode propagation in part 2 of the converter. This TE30 mode is then transmitted to part 4 through

a small section (part 3), which helps to conform the electrical field in such a way as to improve the matching with the TE20 mode in the guide of part 4. As shown in the 3D view (Figure 1b), this last section is a tapered waveguide used to change the height of the waveguide supporting the TE20 mode, with the objective of having a waveguide that supports only the two dominant modes TE10 and TE20 in the Ku-band.

The different parameters (length, width, and height) were finally slightly modified using the HFSS™ 3D-electromagnetic solver to minimize the reflection coefficient and to maximize the conversion ratio between the TE10 and TE20 modes at 15 GHz. The electrical field distribution at 15 GHz in the optimized structure is presented in Figure 2, where it can be clearly seen that the conversion of the TE10 mode to TE20 mode is made at this frequency. The final optimized dimensions are summarized in Table 1. The mode impedances obtained using HFSS are also given in order to show the capabilities of the different waveguides to support the different transmission modes.

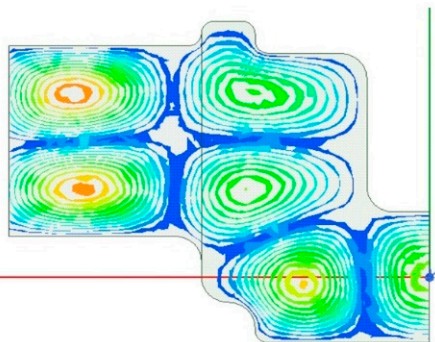

**Figure 2.** Electric field inside the mode converter at 15 GHz.

**Table 1.** Converter dimensions after optimization at 15 GHz.

| Section | Length ($L_i$) (mm) | Width ($a_i$) (mm) | Height ($b_i$) (mm) | $Z_{TE10}$ ($\Omega$) | $Z_{TE20}$ ($\Omega$) | $Z_{TE30}$ ($\Omega$) |
|---------|------------|------------|------------|------------|------------|------------|
| 1 | 7 | 15.8 | 7.9 | 300 | - | - |
| 2 | 12.5 | 34 | 7.9 | 113 | 133 | 237 |
| 3 | 6 | 35 | 7.9 | 109 | 127 | 205 |
| 4 | 25.5 | 22 | 4.14 | 98 | 209 | - |

A back-to-back converter (TE10-TE20-TE10), shown in Figure 3, was then manufactured by DMLS using a ProX300 machine from 3D Systems Ltd., Valencia, CA, USA. The AlSi7Mg0.7 aluminum alloy was used to print the structure. The geometric parameters used as input for the DMLS process are the optimized parameters presented in Table 1. The orientation of the prototypes on the printing plate during DMLS was chosen to achieve the best possible printing quality. As a post-process, the printed parts underwent heat treatment to relax the first macroscopic residual stresses that may appear after the laser sintering operation. Flanges were polished in order to avoid possible air gaps during the RF measurements.

The back-to-back converter was measured using a standard calibration kit and vector network analyzer (VNA) in the Ku-band. The comparison between simulated and measured scattering parameters is presented in Figure 4. A good agreement between HFSS™ results and the measurements with a 50 MHz shift can be observed. This 50 MHz frequency shift (around 0.3% of the central frequency) toward a lower frequency is due to a slight increase in all dimensions, particularly for waveguide width, which results from the DMLS process. Retro-simulations demonstrate that using a scaling factor of 1.004 compared with initial simulation permits to retrieve the measurement curves, as can be seen with the green curve in Figure 4.

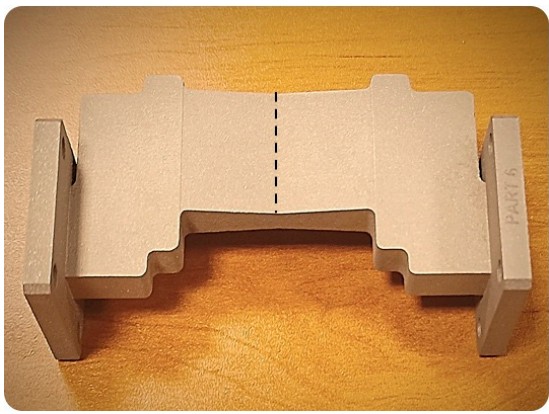

**Figure 3.** DMLS back-to-back TE10–TE20–TE10 mode converter.

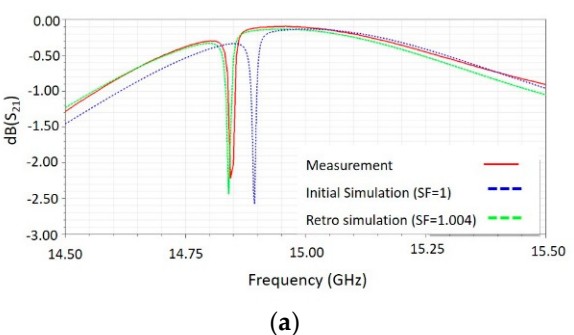

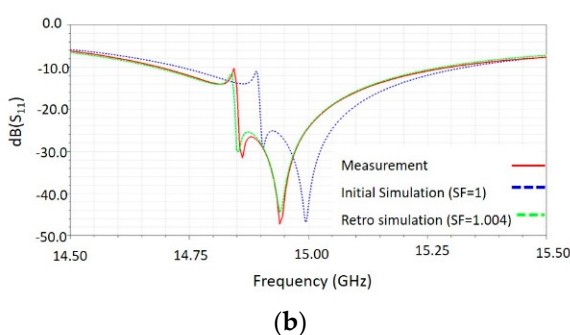

| (**a**) | (**b**) |

**Figure 4.** Simulated and measured S-parameters of the TE10-TE20-TE10 back-to-back converter. (**a**) Transmission coefficient $|S_{21}|$ and (**b**) reflection coefficient $|S_{11}|$.

The measured magnitude of the transmission coefficient $|S_{21}|$ of the double converter at 15 GHz is 0.1 dB and the magnitude of the reflection coefficient $|S_{11}|$ is lower than $-20$ dB for both the simulation and RF measurement results. The DMLS process, therefore, has a very slight impact on the predicted performances.

Both simulated and measured transmission coefficients show a notch at 14.89 GHz for the simulated results and at 14.80 GHz for measurements, respectively (Figure 4). This notch corresponds, at this particular frequency, to poor TE20 excitation, as illustrated in Figure 5a, whereas at 15 GHz, the conversion from TE10 to TE20 is well established, as shown in Figure 5b. This phenomenon is due to the back-to-back converter geometry because it does not appear when only the TE10 to TE20 converter is considered, as observed during the design of the single converter. This will be examined and illustrated in the next section.

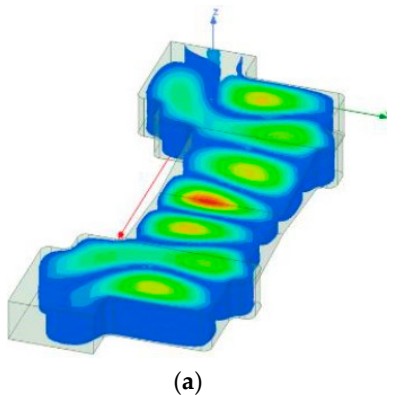

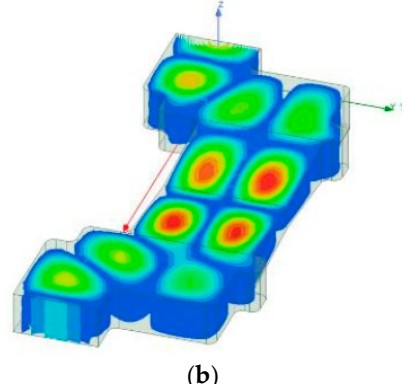

| (**a**) | (**b**) |

**Figure 5.** Electrical field in the back-to-back mode converter at two frequencies; (**a**) at 14.89 GHz and (**b**) at 15 GHz.

These results, therefore, prove that the design of the converter and its manufacture using the DLMS process are suitable for feeding a slotted waveguide antenna.

## 3. Design of a Slotted Waveguide Antenna

### 3.1. Design of a Slot Waveguide Antenna with 16 Slots Excited by TE20 Mode

Using the TE20 mode, the feeding network for a novel 16-slot waveguide antenna was developed. The radiating element is a slotted waveguide supporting the TE20 mode, with short-circuit terminations at each extremity. It is fed at its center by the waveguide using the TE20 mode.

The radiating slots were positioned to cut current lines on metal surfaces in the same way as for a TE10 SWA. Then, to respect a distance of $\lambda g/2$ between $\lambda/2$ long slots on the longitudinal axis of the guide, the positions of slots were alternately placed near the center of the guide and near its walls. As shown in Figure 6 after tuning, the intervals between the closest adjacent radiating slots ($e_1$) and between the farthest ones ($e_2$) are 7.4 mm and 15.6 mm, respectively. The length of the radiating slots is 9.4 mm and their width is 1.8 mm. The total length of SWA ($L_{SWA}$) is 130 mm and the width ($W_{SWA}$) is 25 mm.

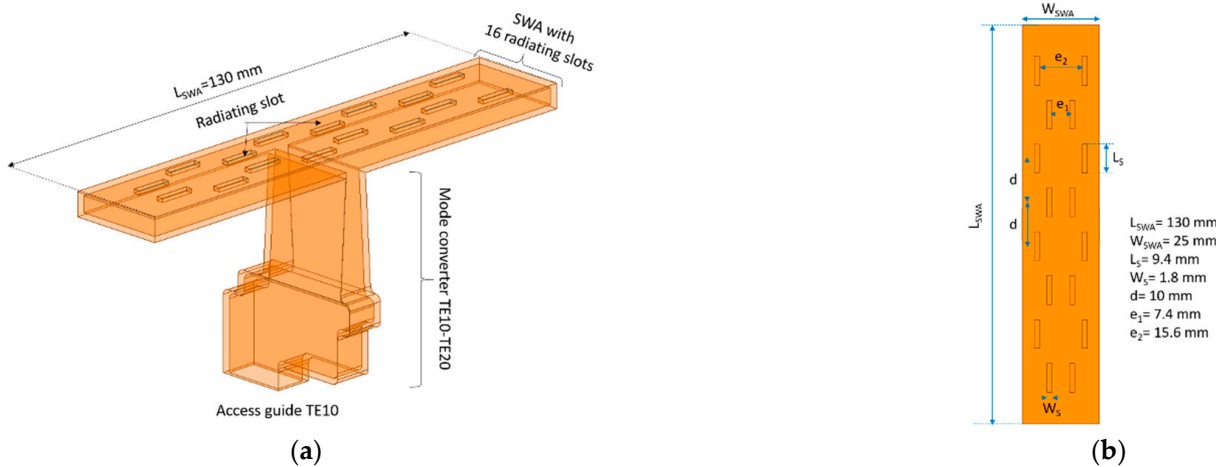

**Figure 6.** Sixteen-slot waveguide excited by the TE20 mode; (**a**) 3D view and (**b**) upper view.

To confirm that the notch at 14.89 GHz present in the parameter $|S_{21}|$ of the back-to-back mode converter has no effect on the designed antenna, the realized gains simulated at 14.89 GHz and 15 GHz are presented in Figure 7. Only small differences can be observed in both plane E and H and no reduction of the maximum gain.

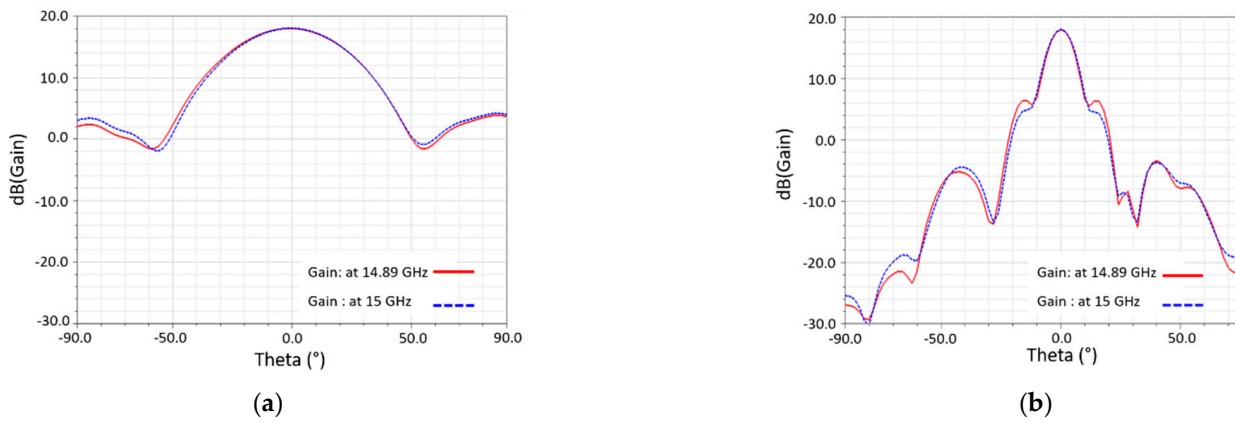

**Figure 7.** Simulated gain pattern at 14.89 and 15 GHz of the 16-slot waveguide antenna: (**a**) E-plane and (**b**) H-plane.

A comparison between the measured and HFSS™ simulated S-parameters is shown in Figure 8. The magnitude of the reflection coefficient $|S_{11}|$ is lower than $-10$ dB over a bandwidth of 700 MHz. A 0.2 GHz frequency shift is observed between the measurement and simulation results, which is due to the dimension variations resulting from the DMLS process. In accordance with the DMLS manufacturing tolerances, a 16-slot waveguide antenna with a scale factor (SF) of 1.01 (SF = 1%) shows a good agreement between the simulated and measured responses (Figure 8). One can note that the notch seen on the back-to-back converter is not yet observable in the antenna behavior, which confirms the hypothesis stated above.

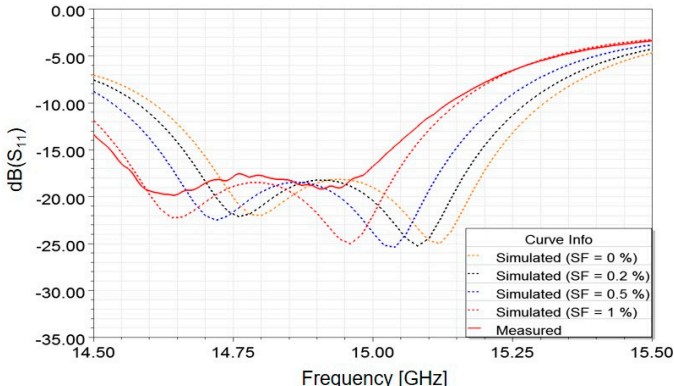

**Figure 8.** Comparison between simulations and measurement of the reflection coefficient $|S_{11}|$ of the 16-slot waveguide antenna for different scale factors (SFs).

Regarding the radiation patterns in the E- and H-planes presented in Figure 9 measured and simulated at 15 GHz, a good agreement is obtained between the EM simulation and measurement results. Indeed, a realized gain of 18 dB is measured in the H-plane, with a sidelobe level 23 dB lower than the main lobe. The cross-polarization level is 30 dB lower than the co-polarization level in both the E- and H- planes.

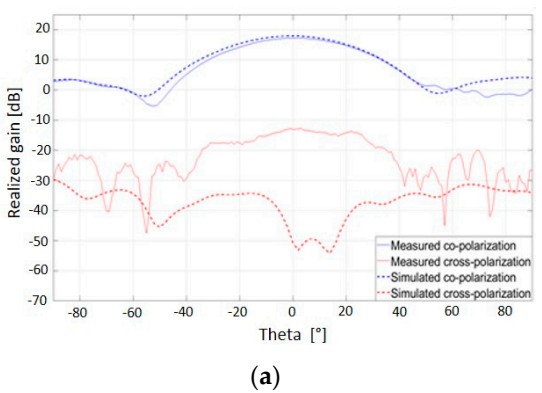 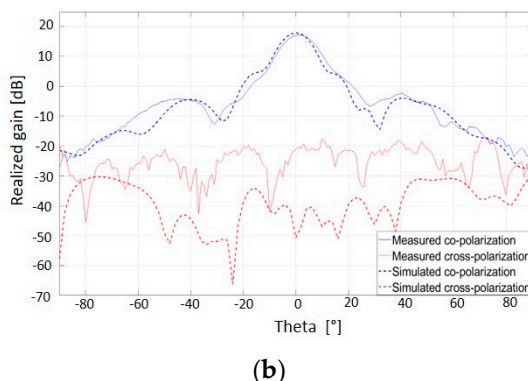

(**a**)  (**b**)

**Figure 9.** Measured and simulated co- and cross-polarization radiation patterns at 15 GHz of the 16-slot waveguide antenna: (**a**) E-plane and (**b**) H-plane.

Based on these results, which confirm the possibility to use the TE20 mode to excite a SWA, an RF power network distribution using the TE20 propagation mode was then designed for a 64-slot waveguide antenna. This is presented in the next section.

### 3.2. Design of the Power Distribution Network of a 64-Slot Waveguide Antenna

The power distribution network propagating the TE20 mode is shown in Figure 10. The first part of this power distribution network consists of the TE10 to TE20 mode converter presented above. Its purpose is to enable the measurement of RF performances such as the reflection coefficient and radiation patterns. This approach allowed us to use only

one mode converter, then the whole power distribution network operates using the TE20 propagation mode, resulting in a simplified design process. The first power divider makes it possible to separate the path in order to excite the 16-slot SWAs n°2 and n°4 on one side and n°1 and n°3 on the other side, as shown in Figure 10a. To ensure balanced in-phase signals on all partial SWA (n°1 to n°3), an electrical length difference corresponding to 2kπ radians (k ∈ ℕ) is introduced according to the TE20 mode on the path connected to n°1 and n°3. Another power divider is placed at each end of both these paths to obtain four different outputs. Then, H-plane bends are used to couple signals to each 16-slot waveguide antenna in the same location, as shown in Figure 10c. The complete power distribution network was simulated and optimized using HFSS™ software. At around 15 GHz, the proposed power distribution network aims to distribute an identical energy supply for each 16-slot SWA, with the lowest possible reflection coefficient. The visualization of the electric field in the distribution network at 15 GHz confirms the propagation of the TE20 mode, as shown in Figure 10b.

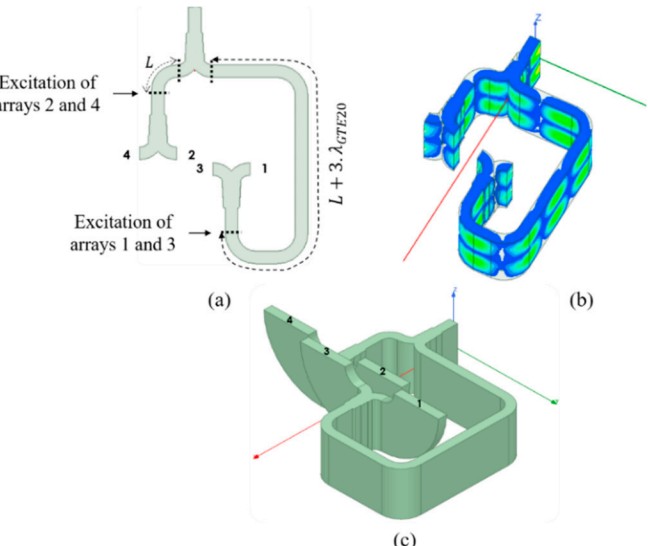

**Figure 10.** Design of the feeding network. (**a**) Top view. (**b**) 3D plot of the electrical field. (**c**) 3D view with coupling bends 1,2,3,4.

The complete novel antenna array was designed according to the following steps. First, the relative working bandwidth is limited to 6% around 15 GHZ. The four 16-slot SWAs were placed next to each other by introducing offsets in order to make space for bends in the E-plane, as shown in Figure 11. At this stage, the interest of using a TE20 mode can be seen. Its cross-section dimensions are of critical importance and must allow the energy to be coupled to each of the 16-slot waveguide antenna arrays. Indeed, the shift between the phase centers of each antenna depends strongly on the width of the waveguide (the minimum possible value is 22 mm).

In the same way as for the SWA with 16 slots, the radiating slots were placed to cut current lines on the metallic part of the guide. The positioning of the radiating slots was chosen by placing two adjacent slots close to the center of the TE20-mode waveguide, and then the two next adjacent slots in the direction of propagation near the edges of the TE20-mode waveguide, as seen in Figure 11b. This alternate positioning of the slots allows the space between them to be reduced, and thus limits the sidelobe levels.

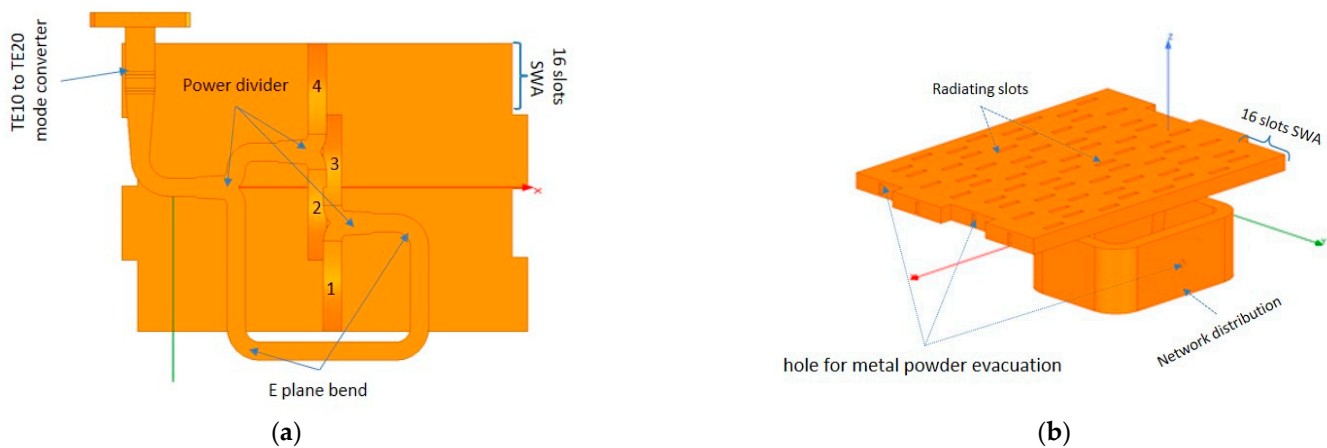

**Figure 11.** Design of the SWA array. (**a**) Bottom view with the bottom access 1,2,3,4 of the four 16 slots SWA and (**b**) 3D view.

The proposed metallic SWAs were printed using the DMLS manufacturing process. The antennas were manufactured layer by layer from a metallic powder bed using a Prox DMP 300B 3D printer. The thickness of each layer was 40 µm. Typically, the accuracy of the Prox DMP 300B printer in the horizontal plane was equal to ± 0.1 mm. The printed material was the aluminum alloy AlSi7Mg0.6, which needs a protective argon atmosphere. During the design step, holes were positioned where the electrical field was weak to easily evacuate the powder trapped inside the structure at the end of the DMLS manufacturing process, as shown in Figure 11b.

The orientation of the antennas during the 3D printing process was chosen to ensure the best accuracy of the radiating slots and to respect DMLS recommendations.

Figure 12 shows the choice of orientation in the printing chamber for the SWA array. The antenna was printed with the plates with the radiating slots oriented to be supported and to respect a 45° orientation to build the powering waveguide network. This printing orientation helps to control the dimensions of the slots because mechanical machining is needed to remove the support on the radiating surface. A good surface state of the radiating plane is thus obtained and the radiating slot dimensions can be measured and, if necessary, be rectified using classical machining.

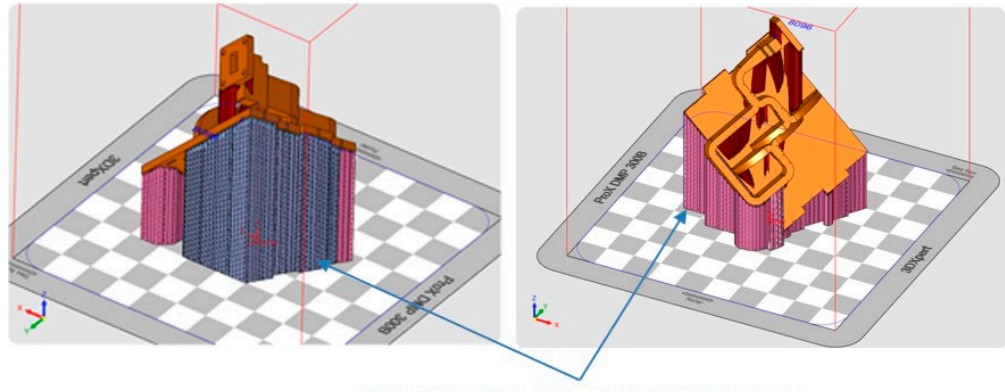

**Figure 12.** Orientation of the SWA array in the printing chamber and placement of the support structures.

Figure 13 shows a photograph of the antenna after the DMLS manufacturing process and application of the required post-processing (heat treatment, removal of the support structures, sandblasting, and polishing of the connection flange).

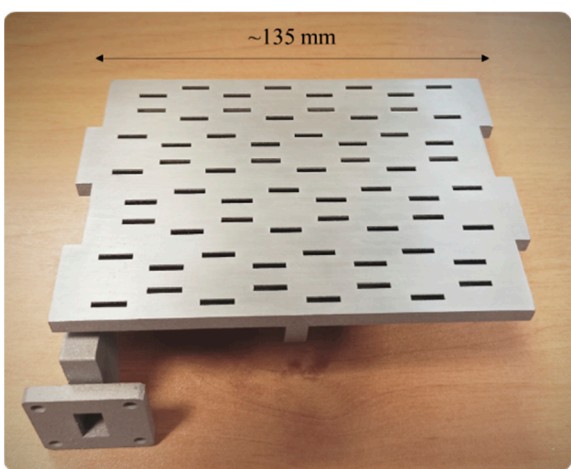

**Figure 13.** Photograph of the SWA array realized by the DMLS process.

Figure 14 presents a comparison between the simulated and measured reflection $S_{11}$ coefficients. The measured magnitude of $S_{11}$ at 15 GHz is lower than $-12.6$ dB. The difference between the simulated and measured responses is due to some inaccuracies of the DMLS manufacturing process, as DMLS 3D printing gives larger dimensions than those designed and this causes shifts in the RF response toward low frequencies.

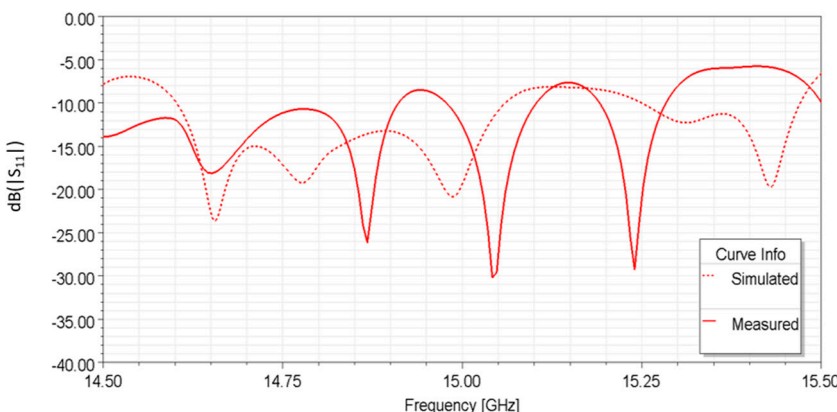

**Figure 14.** Simulated and measured reflection coefficients of the printed SWA array.

The radiation characteristics of the printed SWAs were measured in an anechoic chamber. The radiation patterns were measured in both the E- and H-planes in two independent tests. The experimental results at 15 GHz are compared with the simulated ones in Figure 15. It can be seen that the simulated and measured radiation patterns are in good agreement. The on-axis realized gain is 22.26 dB, with a sidelobe level lower than 10.1 dB for both planes. These results are similar to those presented in [16] with less complex powering.

The difference between the electromagnetic simulation results and measurements of the cross-polarization level on the SWA axis is mainly due to a misalignment between the antennas (test antenna and measured antenna) during the measurement procedure. Nevertheless, the cross-polarization remains below 20 dB in both the E- and H-planes. These results confirm the concept of using a monolithic DMLS process to design complex RF multifunctions such as a TE20 mode to excite an SWA (64 slots) with an associated RF power network distribution.

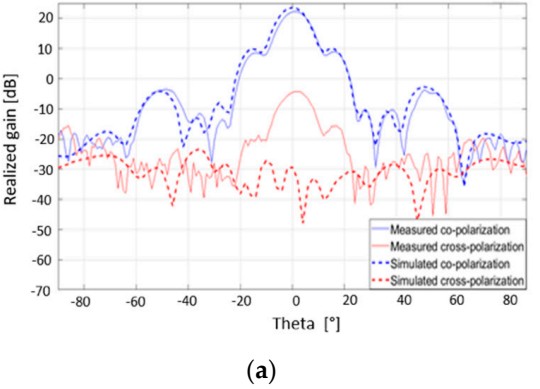

(**a**)

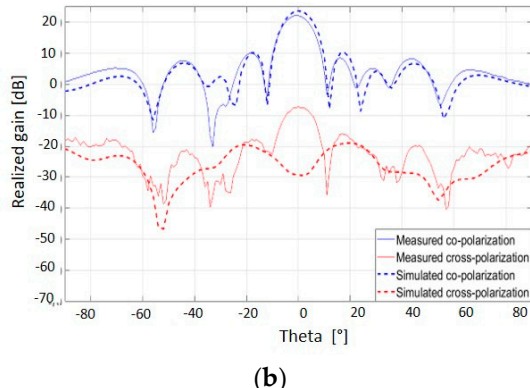

(**b**)

**Figure 15.** Radiation pattern (gain) of the SWA: comparison of simulated and measured co- and cross-polarization at 15 GHz. (**a**) E-plane pattern and (**b**) H-plane pattern.

## 4. Conclusions

With a view to building an SWA array for the Ku-band using the DMLS 3D printing process, notably avoiding sensitive elements like irises or magnetic posts inside the structure and exciting more slots, a new approach was developed and designed to feed the array. To this end, the use of the TE20 propagation mode was investigated and a power distribution network supporting this propagation mode is proposed that makes it possible to excite the $4 \times 16$ slotted waveguide antenna. First, a TE10 to TE20 mode converter was developed, and then a 16-slot antenna using the TE20 mode. Finally, a Ku-band SWA array was designed and then manufactured using the DMLS process, taking particular care about the printing orientation of the antenna. The RF measurements show that the on-axis realized gain is 22.26 dB with a measured return loss lower than $-12.6$ dB at 15 GHz. These studies confirm that the power distribution network performs well and, therefore, that the power distribution design is suitable for building a monolithic SWA array in one step using the DMLS process.

Through this study, we have demonstrated that using a non-conventional TE20 propagation mode to realize the powering network of slotted waveguide arrays is a solution to overcoming some DMLS manufacturing imperfections or defects while keeping the benefits of a monolithic process for complex RF functions. One of the future perspectives of this work consists of using this approach to design a compact monopulse antenna system with an integrated comparator network in a manner compatible with the DMLS process.

**Author Contributions:** Conceptualization, S.A.; Funding acquisition, T.M.; Supervision, R.A., A.-C.A., A.E.O., C.Q., K.R. and T.L.G.; Writing—original draft, Y.C. All authors have read and agreed to the published version of the manuscript.

**Funding:** This research received no external funding.

**Acknowledgments:** The authors would like to thank Thales T3DM Ltd. for manufacturing the antenna, as well as Thales LAS France SAS. The authors would also like to thank the TECHYP platform (the High-Performance Computing Cluster of Lab-STICC), where the devices could be simulated. Finally, the authors would like to thank ENSTA-Bretagne engineering school for the use of their anechoic chamber and particularly Fabrice Comblet for his help. This publication is supported by the European Union through the European Regional Development Fund (ERDF) and by the Ministry of Higher Education and Research and Brest Métropole, Brittany, through the CPER Project SOPHIE / STIC & Ondes. This work was carried out as part of the CIFRE-France/Morocco program.

**Conflicts of Interest:** The authors declare no conflict of interest.

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
