# Peer review of "Design of a Slotted Waveguide Antenna Based on TE20 Mode in Ku-Band Suitable for Direct Metal Laser Sintering"

_electronics, doi:10.3390/electronics11132079_

Round 1
Reviewer 1 Report
The paper titled " Design of a Slotted Waveguide Antenna based on 2 TE20 Mode in Ku-band suitable for Direct Metal La-3 ser Sintering " proposed a
design of a Ku-band slotted waveguide antenna (SWA), using a 3D direct metal laser sintering 13 (DMLS) process. The advantages and drawbacks of the 3D direct metal laser sintering based processing method is considered during the antenna design. The topic is interesting. However, some issues should still be addressed.
1. During the rectangular waveguide TE10 to TE20 mode converter design, the geometric parameters of the shown in Table I should be in a range, since the machining error exists. Hence, the impedance results can be predicted for a fabricated converter. In addition, how are the impedance results shown in Table I obtained, calculated, simulated or measured?
2. The resonant frequency of the converter shifts about 50MHz, Please give a brief simulation to illustrate the main factor for this phenomenon.
3. For the DMLS process, the accuracy is different for each dimension. The accuracy is usually poor in the dimension of the support direction. Please analyze the tolerance to the machining error of the antenna design especially in the support direction.
Author Response
Thank you for your review.
Please find below our responses to your constructive remarks.
- During the rectangular waveguide TE10 to TE20 mode converter design, the geometric parameters of the shown in Table I should be in a range, since the machining error exists.
You are right, the geometric parameters after DMLS printing are different than the ones used as input. In table I, the presented geometric parameters are the ones used as input data for DMLS process. They are issued from optimization and it is difficult to know a priori the geometry change. To clarify this point, we have modified the sentence line 108 as:
“The geometric parameters used as input for DMLS process are the optimized presented Erreur ! Source du renvoi introuvable. »
We have also modified the legend of table 1.
- Hence, the impedance results can be predicted for a fabricated converter. In addition, how are the impedance results shown in Table I obtained, calculated, simulated or measured?
We agree with you, and the precision has been added in the text.
“The mode impedances obtained using HFSS”
- The resonant frequency of the converter shifts about 50MHz, Please give a brief simulation to illustrate the main factor for this phenomenon.
A retro-simulation with a scaling factor of 1.004 (0.4 %) in all dimensions (x,y,and z) can explain the 50 MHz shift. Figure 4 has been modified with the retro-simulation results
The following sentence have also been added.
“Retro-simulations demonstrate than using a scaling factor of 1.004 compared to initial simulation permit to retrieve the measurement curves as it can be seen with the green curve in figure 4”
- For the DMLS process, the accuracy is different for each dimension. The accuracy is usually poor in the dimension of the support direction. Please analyze the tolerance to the machining error of the antenna design especially in the support direction.
As the radiating pattern of the SWA antenna greatly depend of radiating slot dimensions, our choice of printing orientation is to place the support on this radiating surface, because of the need of mechanical machining to remove the support.
A comment has been added in the sentence.
“A good surface state of the radiating plane is thus obtained and the radiating slot dimensions can be measured and if necessary be rectified using classical machining.”
Thanks again for your interesting comments.
Reviewer 2 Report
COMMENTS:
The submitted manuscript presents the design of a slotted waveguide antenna (SWA) with 16 slots, a TE10 to TE20 mode converter, and an array of four SWAs including its feed network. These components were printed using an aluminum alloy and good electrical performance was observed with a return loss better than 10 dB over almost the entire tested band and the patterns show low cross-polarization level, controlled sidelobe levels, and good symmetry. The results of the work are certainly of interest to the readers of Electronics. Some comments are addressed to the authors in order to improve the quality of the publication.
1) Figure 4(b) and Figure 13 present the reflection coefficient and not the return loss. The proper definition of return loss can be found in T. S. Bird, “Definition and misuse of return loss,” IEEE Antennas and Propagation Magazine, vol. 51, no. 2, pp. 166–167, Apr. 2009, for example. Check line 264 for the same reason.
2) If possible, in addition to the reflection coefficient of Figure 7, the patterns in the notch frequency of Figure 4 could be presented to confirm it does not affect the antenna performance.
3) Are the patterns in Figure 8 plotted at 15 GHz? This data should be included in the text.
4) The specification for the frequency band of the SWA array should be described in the text. Otherwise, it is difficult to evaluate the response in Figure 13, since for some intervals the return loss is below 10 dB.
5) The authors could add more information to support the use of the TE20 mode to complement the explanation given in line 195.
Author Response
Thank you for your review.
Please find below our responses to your constructive remarks.
- Figure 4(b) and Figure 13 present the reflection coefficient and not the return loss. The proper definition of return loss can be found in T. S. Bird, “Definition and misuse of return loss,” IEEE Antennas and Propagation Magazine, vol. 51, no. 2, pp. 166–167, Apr. 2009, for example. Check line 264 for the same reason.
You are right and the y axis of the figure has been corrected.
- If possible, in addition to the reflection coefficient of Figure 7, the patterns in the notch frequency of Figure 4 could be presented to confirm it does not affect the antenna performance.
To proof that there is no effect of the notch in the S21 parameter at 14.89 GHz as shown in figure 4 , we have added a figure presenting the simulated gain at the two frequencies 14.89 GHz and 15 GHz.
The following sentences have also been added.
“To confirm that the notch at 14.89 GHz present in the parameter |S21| of the back to back mode converter, has no effect on the designed antenna, the realized gain simulated at 14.89 GHz and 15 GHz are presented in Figure 6. Only small differences can be observed in both plane E and H and no reduction of the maximum gain. “
3) Are the patterns in Figure 8 plotted at 15 GHz? This data should be included in the text.
You are right, this information has to be given. The precision has been added line in text and also in the figure legend.
4) The specification for the frequency band of the SWA array should be described in the text. Otherwise, it is difficult to evaluate the response in Figure 13, since for some intervals the return loss is below 10 dB.
The following sentence has been added to specify the interesting bandwidth that is small (6%).
“The complete novel antenna array was designed according to the following steps. First the relative working bandwidth is limited to 6 % around 15 GHZ.”
5) The authors could add more information to support the use of the TE20 mode to complement the explanation given.
To complete the design process of power distribution network the sentence has been added .
“This approach allowed us to use only one mode converter, then all the power distribution network operates using the TE20 propagation mode, resulting in a simplify design process.”
Thanks again for your interesting comments.
Reviewer 3 Report
Presented article is devoted to implementation of direct metal laser sintering (DMLS) for microwave antenna array manufacturing.
Using the DMLS for such an application is known, however authors propose a technology-oriented design of antenna array with good agreement between simulations and measurements. Thus, this work may be interesting for researchers in the field.
I have several remarks and suggestion to improve the readability of the article.
1. Since it is possible to use subscripts in the article's text, it would be better to use it for stoichiometric coefficients, see lines 102, 212.
2. Figure 6 has poor quality. Authors used dashed lines, dashed arrows and solid lines for illustration in a small area of the image, thus, it requires additional efforts to read the picture. At least these lines should be more contrast to the background for clearness. In the figure caption, the word "exited" capitalised for no reason.
3. It would be correct to write |S11| instead of S11, see line 151.
4. The y-label is incorrect in figure 7. Please fix it in accordance with x-label.
5. The expression "n \cross WLTE20" is not clear. These notations are presented in figure 9 only and without description.
6. There is some solid line below x-label in figure 13, please remove it.
7. I suppose that Dr. Gregory Peter Le Sage is a single person and U.S. Government is his affiliation, but not coauthor. Please check ref. 8.
8. Authors’ affiliations are not complete. It should include "complete address information including city, zip code, state/province, and country."
Author Response
Thank you for your interesting remarks and suggestion.
Please find below our responses.
- Since it is possible to use subscripts in the article's text, it would be better to use it for stoichiometric coefficients, see lines 102, 212.
AlSi7Mg0.7 correspond to the denomination of aluminum alloys with 7% of Si and 0.7% of Mg. It is not a stoichiometric coefficient
We refer to steel number website.
http://www.steelnumber.com/en/steel_alloy_composition_eu.php?name_id=1224
- Figure 6 has poor quality. Authors used dashed lines, dashed arrows and solid lines for illustration in a small area of the image, thus, it requires additional efforts to read the picture. At least these lines should be more contrast to the background for clearness. In the figure caption, the word "exited" capitalised for no reason.
For more readability, the Figure 7 (in new version) has been completed with an upper view of SWA. The word exited have been corrected.
- It would be correct to write |S11| instead of S11, see line 151.
This has been corrected
- The y-label is incorrect in figure 7. Please fix it in accordance with x-label.
Sorry I do not really understand your remark.
- The expression "n \cross WLTE20" is not clear. These notations are presented in figure 9 only and without description.
The figure 9 has been modified by L+3lgTE20
- There is some solid line below x-label in figure 13, please remove it.
The correction has been done.
- I suppose that Dr. Gregory Peter Le Sage is a single person and U.S. Government is his affiliation, but not coauthor. Please check ref. 8.
You are right, the correction has been done
- Authors’ affiliations are not complete. It should include "complete address information including city, zip code, state/province, and country."
All author’s affiliations have been completed
Thank you again for all your suggestions and remarks.
Round 2
Reviewer 1 Report
No further comments.